# *STH-net*: a soil monitoring network for process-based hydrological modelling from the pedon to the hillslope scale

Edoardo Martini[1,2], Matteo Bauckholt[2], Simon Kögler[a], Manuel Kreck[2], Kurt Roth[1], Ulrike Werban[2], Ute Wollschläger[3], Steffen Zacharias[2]

[1] Institute of Environmental Physics, Heidelberg University, Heidelberg, 69120, Germany
[2] Dept. Monitoring and Exploration Technologies, Helmholtz Centre for Environmental Research GmbH - UFZ, Leipzig, 04318, Germany
[a] formerly at: Dept. Monitoring and Exploration Technologies, Helmholtz Centre for Environmental Research GmbH - UFZ, Leipzig, 04318, Germany
[3] Dept. Soil System Science, Helmholtz Centre for Environmental Research GmbH - UFZ, Halle (Saale), 06120, Germany
*Correspondence to*: Edoardo Martini (emartini@iup.uni-heidelberg.de)

**Abstract.** The *Schäfertal hillslope* site is part of the TERENO Harz/Central German Lowland Observatory and its soil water dynamics are being monitored intensively as part of an integrated, long-term, multi-scale and multi-temporal research framework linking hydrological, pedological, atmospheric and biodiversity-related research to investigate the influences of climate and land use change on the terrestrial system. Here, a new soil monitoring network, indicated as *STH-net*, has been recently implemented to provide high-resolution data about the most relevant hydrological variables and local soil properties. The monitoring network is spatially optimized, based on previous knowledge from soil mapping and soil moisture monitoring, in order to capture the spatial variability of soil properties and soil water dynamics along a catena across the site as well as in depth. The *STH-net* comprises eight stations instrumented with time-domain reflectometry (TDR) probes, soil temperature probes and monitoring wells. Furthermore, a weather station provides data about the meteorological variables. A detailed soil characterization exists for locations where the TDR probes are installed. All data are measured at a 10-minutes interval since January 1st, 2019. The *STH-net* is intended to provide scientists with data needed for developing and testing modelling approaches in the context of vadose-zone hydrology at spatial scales ranging from the pedon to the hillslope. The data are available from the EUDAT portal (https://b2share.eudat.eu/records/82818db7be054f5eb921d386a0bcaa74) (Martini et al., 2020).

## 1 Introduction

Soils are embedded in the environment, coupled to vegetation and atmosphere at the land surface and to groundwater at its lower end. This coupling gives rise to a suite of physical, chemical, and biological dynamics most of which are highly non-linear and varying in time and space. Soils provide crucial ecosystem functions such as water storage and filtering, food and other biomass production, recycling of carbon and nutrients, biological habitat and gene pool, physical and cultural heritage, source of raw materials and platforms for human life (United Nations, 2014; Vereecken et al., 2016). Soils are widely distributed on the Earth surface. Flow and transport processes in unsaturated soils occur predominantly in the vertical direction, with the gravity force playing a major role, as abrupt changes in soil properties due to soil horizons and layers are typically

more significant than those in the lateral direction, and because of the strong coupling between soil, vegetation, and
atmosphere. Therefore, despite the relevance of soils for global phenomena, the relevant soil processes are rather local. Here,
one aspect that complicates the picture is the heterogeneity of soil properties. Another one is the non-linearity of soil processes.
In order to address effectively this complexity, state-of-the-art experimental approaches must be coupled to numerical models
for the comprehensive representation of the system properties, states and fluxes so that the hydrological system can be better
understood.
Recently, Vogel (2019) provided a comprehensive discussion about the scales and scaling issues in the context of soil
hydrological research and noted the need for looking at small-scale soil properties (i.e., at the pedon scale, at which soil physics
is capable of describing states and fluxes with sufficient accuracy) as a necessary step towards understanding and summarizing
the processes at larger scales. In this respect, the author stresses the need for a two-steps approach based on the accurate
description of the soil water dynamics at the pedon scale and accounting for the spatial patters of functional soil types that
constitute the landscape, including the vertical stratification of soil hydraulic properties and structural attributes. However, the
author remarks that high-resolution measurements of the relevant states and properties cannot be achieved at the larger scale
(i.e., catchment, the typical scale of application of hydrological research). In this context, the intermediate scale of hillslopes
is crucial for linking the detailed process understanding to larger scale dynamics, recognizing hillslopes as key landscape
features that organize water availability on land (Fan et al., 2019). In this respect, coupling state of the art hydrological
modelling approaches with high-resolution subsurface characterization can lead to an accurate quantification of the soil water
dynamics in the vadose zone (Vereecken et al., 2015).
The physical description of the small-scale water movement through the soil's porous structure is typically achieved using the
Richards equation. However, the detailed description of the material properties is needed and cannot be fully resolved by direct
sampling. Thus, inverse modelling can be a powerful tool for the estimation of the soil hydraulic parameters (e.g., Vrugt et al.,
2008), including the recent developments in data assimilation approaches (e.g., Bauser et al., 2016, 2020; Botto et al., 2018).
These require dense (in the direction of the dominant flow, typically orthogonal to the soil surface) measurements of soil water
content with high temporal resolution and of high quality. Furthermore, *in situ* sensors can experience all the processes
affecting the measured state variables in their natural environment (Wollschläger et al., 2009), which is an important advantage
with respect to sample-based determinations from the laboratory.
The performances of hydrological models can be improved by various measured data with high spatial and temporal resolution
(Clark et al., 2017). Bronstert (1999) highlighted the importance of linking experimental knowledge to the experience gained
from numerical modelling applications as a very valuable synergistic combination. Technological advances in our ability to
measure soil hydrological states efficiently at the hillslope scale and beyond are one possible way to gain the much-needed
improved understanding of processes that challenge the comprehensive understanding of field-scale hydrology.
In the research framework of the TERENO Harz/Central German Lowland Observatory, the *Schäfertal hillslope* represents a
benchmark site for developing and testing the integration of state-of-the-art monitoring techniques with advanced modelling
approaches. This offers the opportunity to gain a more detailed understanding of processes and to quantify and predict water
and matter fluxes at nested spatial scales in the context of climate and land use change. Specifically, the approach followed at
the site accounts for the soil spatial variability through detailed soil mapping and is designed to provide *in situ* data with high
temporal resolution and dense coverage in the vertical direction, about the soil water dynamics in the vadose zone and of its
boundary conditions. With this design tailored to the needs of vadose zone modelling, we aim to provide physical models with
ideally all the data needed for quantifying and predicting the soil water fluxes at spatial scales ranging from the pedon to the
hillslope scale, with important implications, in terms of methodological advance and process understanding, for catchment-
scale processes.
Here, we present the first 21 months of the comprehensive dataset measured by the monitoring network *STH-net*, recently
implemented at *Schäfertal Hillslope* site, part of an intensive hydrological observatory. The data set includes hourly time series
of the meteorological forcing, soil water content measured *in situ* at different locations and at multiple soil depths along a
hillslope transect, soil physical and physicochemical properties.
**2 Site description**
The Schäfertal experimental site is a small headwater catchment (1.44 km$^2$) located in the Lower Harz Mountains, in Central
Germany (51°39' N, 11°3' E). Environmental research at the Schäfertal catchment was initiated at the end of the 1960s with
the implementation of a hydro-meteorological station (Reinstorf et al., 2010) and the infrastructure has continuously been
expanded since then. Since 2010, the Schäfertal catchment is one of the highly instrumented intensive research sites within the
TERENO Harz/Central German Lowland Observatory (Zacharias et al., 2011, Wollschläger et al., 2017). Due to the
geographical settings of the Harz region, the Schäfertal catchment receives only 630 mm of precipitation per year. The average
annual air temperature is 6.9°C, with a sub-continental climate (Reinstorf, 2010). The geology of the catchment is dominated
by Devonian argillaceous shales and greywackes, covered by periglacial sediments (Borchardt, 1982). Near-surface compacted
horizons within the basal layer are known to induce interflow processes in the unsaturated zone (Borchardt, 1982; Gräff et al.,
2009). Dominant soil types in the Schäfertal are Gleysols occurring in the valley bottom as well as Luvisols and Cambisols on
the loess-covered slopes (Ollesch et al., 2005). The slopes of the catchment are intensively used for agriculture, whilst meadows
occupy the valley bottom (Schröter et al., 2015).
Since 2012, a smaller hillslope area named *Schäfertal Hillslope* site, located downstream of the Schäfertal gauging station,
was instrumented for detailed investigations of the hydrological processes in the unsaturated zone. From 2012 to 2017, the
wireless soil moisture monitoring network *SoilNet* has delivered information about the soil water dynamics at three depths
within the unsaturated zone with high spatial coverage. In 2018, the *SoilNet* has been disposed and a new soil monitoring
network, named *STH-net*, has been installed aiming to improve the resolution in the vertical direction at a fewer locations
selected based on the knowledge about the soil spatial variability and soil water dynamics gained from the previous monitoring
experience (see Martini et al., 2015; 2017a; 2017b). The *STH-net* is described in the following sections of this manuscript and
its        data        are        now        available        through        the        data        portal        EUDAT
(https://b2share.eudat.eu/records/82818db7be054f5eb921d386a0bcaa74). The *Schäfertal Hillslope* site includes north- and
south-exposed slopes divided by the creek (*Schäferbach*) in the valley bottom (Fig. 1). In contrast to the slopes upstream,
which are primarily covered by cropland, this grassland transect is used as pasture and is not affected by agricultural practices
except that the grass is mowed typically once per year. The spatial extent of the hillslope is approximately 250 by 80 m and
presents various topographical and pedological features. The slopes are covered by silty loam Cambisols more evolved towards
the footslope, while loam and silty loam stagnic Gleysols occupy the valley bottom. An extensive description of the soil units
mapped at the site is provided in Martini et al. (2015). The *STH-net* is designed to cover the spatial variability of the soil
properties as well as the soil layering with high resolution.

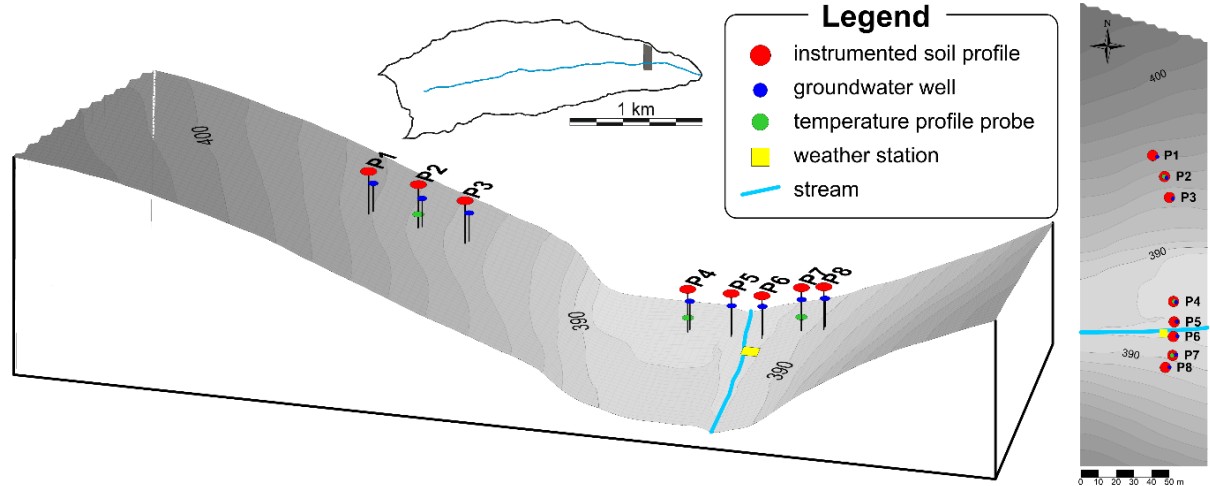


**Figure 1: Spatial map in 3D and aerial view of the *Schäfertal hillslope* site and location of the monitoring stations.**
**3 Monitoring design and measurement techniques**
The *STH-net* comprises eight monitoring stations (named as P1 to P8) arranged along a transect centred within the *Schäfertal*
*Hillslope* site and aligned along the slope direction (Fig.1). The stations P1, P2 and P3 are located within the Northern (i.e.,
South-facing) slope and cover the transition between the soil units STU1 and STU2 described in Martini et al. (2015); the
stations P4 and P5 fall into the valley bottom, i.e., soil unit STU3; P6, P7 and P8 cover the lower part of the Southern (i.e.,
North-facing) slope, i.e., soil unit STU4. Every station features a soil profile instrumented with Time-Domain Reflectometry
(TDR) probes installed every 0.1 m along the vertical direction. A sketch showing the design of a reference monitoring station
is presented in Fig. 2. Each of the instrumented soil profiles located on the hillslopes features a minium of seven TDR probes
installed at the depths of 0.1, 0.2, 0.3, 0.4, 0.5, 0.6 and 0.7 m, whilst an additional probe is installed at P3 at the depth of 0.8 m
and the profiles at P4 and P5 feature additional TDR probes at the depths of 0.8, 0.9, 1.0 and 1.1 m in order to cover the deeper
soils. In a few cases, the depths of the probes were adjusted to avoid installing the TDR probe at or too close to the boundaries
between soil horizons. The exact depth of every TDR probe is reported in the file "STH-net_Soils.txt" and displayed in Fig.

122 3.

At every station, a well instrumented with a piezometer was installed ca. 2 m to the East of the instrumented soil profiles for
monitoring the water level. One station for every topographic unit (i.e., Northern slope, valley bottom and Southern slope) was
further instrumented with sensors measuring the soil temperature at six depths between 0.05 and 1.0 m.
A weather station is located in the centre of the hillslope transect next to the creek.
All measurement systems comprising the *STH-net* collect measurements every 10 minutes, with the only exception of the
water level data which are collected every 2 hours.

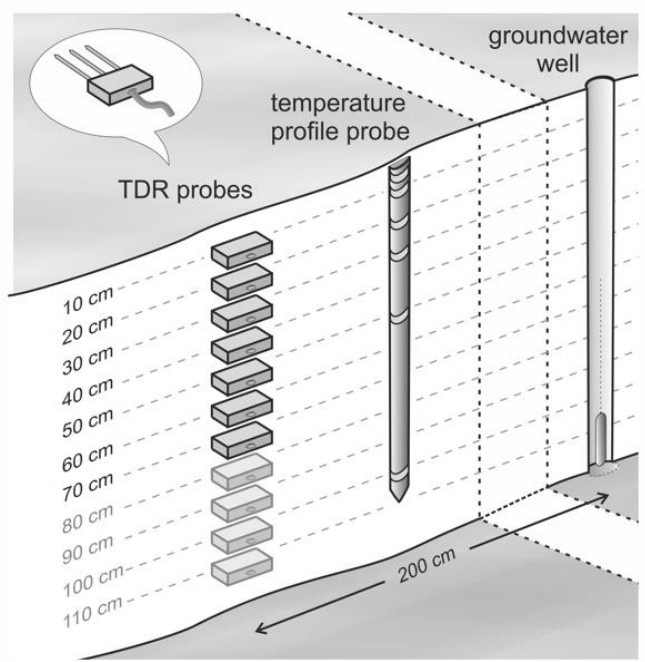


**Figure 2: Sketch of a representative monitoring station of the *STH-net*.**

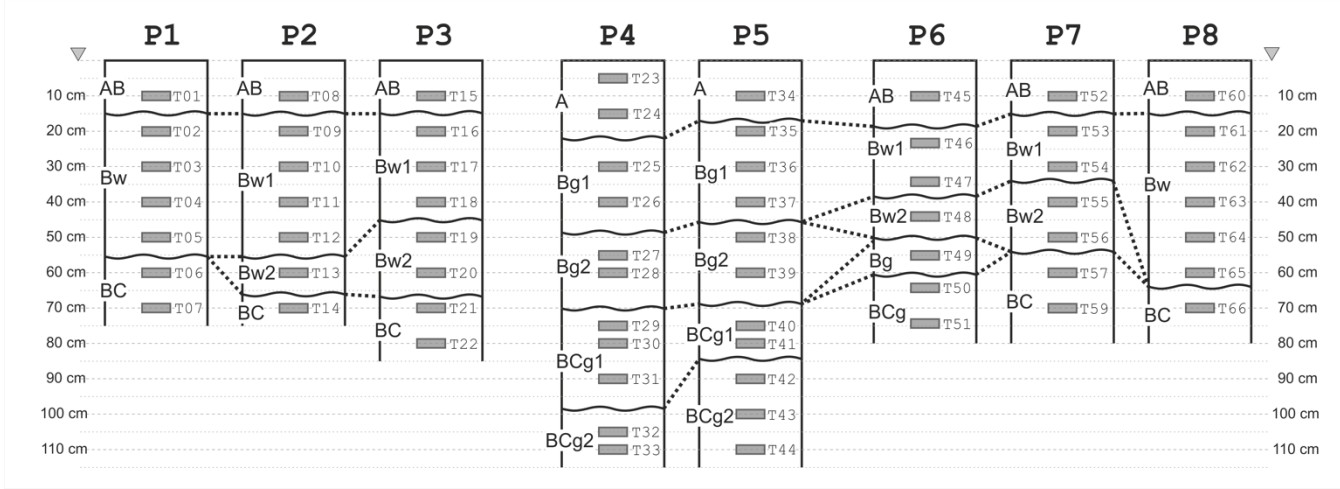

Figure 3: Sketch of the soil profiles (showing the mapped soil horizons according to WRB 2015) and the depth of the TDR probes (see labels).

### 3.1 TDR measurements

The TDR probes are arranged in clusters of 22 probes for the Northern slope and the valley bottom, whilst only 21 probes were installed at the Southern slope, for a total of 65 TDR probes. Each cluster consists of one TDR device (TDR100 for the station North, TDR200 for the stations Valley and South, Campbell Scientific Inc., Logan, UT, United States) and a data logger (CR1000 for the station North, CR6 for the stations Valley and South, Campbell Scientific Inc., Logan, UT, United States). The clusters are powered by extra low voltage cables buried ca. 0.3 m below the ground and cased in HDPE (i.e., high-density polyethylene) tubes and an AGM (i.e., absorbent glass mat) battery capable of supplying the required power in case of power cut-off. Every TDR probe is connected to its station master by a 22-m long low loss coaxial cable, tested to be the optimal length providing good signal quality while enabling enough flexibility in terms of network design. The TDR probes were custom made and have three 0.2 m-long rods. They were calibrated through measurements in air and in water with different salt concentrations for water content and electrical conductivity estimation. The probes were installed horizontally in soil pits which were carefully refilled after the installation. The installation was carried out between June and August 2018 and all the measurements collected until the end of December 2018 were discarded to allow the soil to re-compact naturally during the first rainy season.

From the TDR traces, the dielectric permittivity $\varepsilon$ of the medium is calculated as:

$$\sqrt{\varepsilon} = \frac{\left(\sqrt{\varepsilon_{air}} - \sqrt{\varepsilon_{water}}\right)(t - t_{water})}{t_{air} - t_{water}} + \sqrt{\varepsilon_{air}} \qquad (1)$$

based on the calibration measurements of travel time and dielectric permittivity in air ($t_{air}$, $\varepsilon_{air}$) and water ($t_{water}$, $\varepsilon_{water}$), where $t$ is the travel time estimated for the measured trace. The volumetric water content $\theta$ is calculated according to the complex refractive index model (CRIM) following Roth et al. (1990) as:

$$\theta = \frac{\sqrt{\varepsilon} - \sqrt{\varepsilon_{soil}} - \phi(\sqrt{\varepsilon_{air}} - \sqrt{\varepsilon_{soil}})}{\sqrt{\varepsilon_{water}} - \sqrt{\varepsilon_{air}}} \qquad (2)$$

152    where $\phi$ is the porosity which was calculated from the soil bulk density and $\varepsilon_{soil}$ is set to 4.6. Fig. 4 shows the hourly time

153    series of soil water content. Characteristic differences in the soil water dynamics are evident for the distinct soil profiles and

154    depths to be attributed, e.g., to the differences in soil texture and soil layering or, locally to groundwater dynamics.

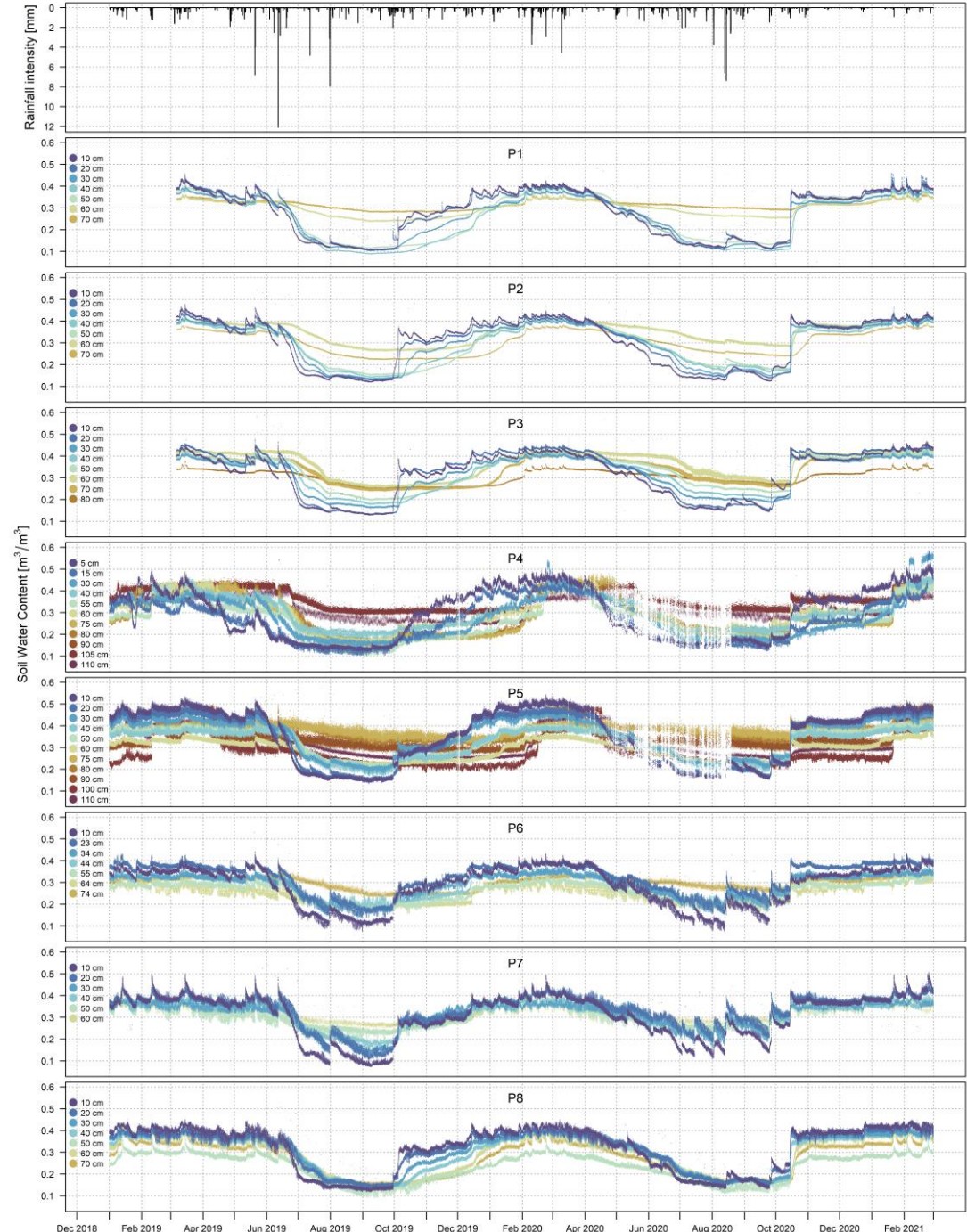

Figure 4: Time series of soil water content data. The plots were made using the data set as it appears in the online archive. The data are plotted using a scientific colour scale from Crameri (2018) chosen according to the principles described in Crameri et al. (2020).

## 3.2 Soil temperature

The stations P2, P4 and P7 are instrumented with one Th3-s soil temperature profile probe (formerly UMS GmbH, Munich, Germany) each, located nearby the instrumented soil profiles (Fig. 2) and connected via SDI-12 to the same data loggers and power supply. The probes consist of six temperature sensors cased inside a tube made of glass-fiber reinforced plastic and placed at the fixed depths of 5, 10, 20, 30, 50 and 100 cm. Soil temperature is measured at the same times as the TDR traces. The measured data are shown in Fig. 5. The influence of the geographical exposure of the slopes is particularly evident, e.g. overall higher temperature and stronger dynamics for the south-exposed slopes compared to the other areas, as well as the strongest dynamics near the surface compared to the deepest sensors. For every temperature profile, the soil temperature values corresponding to the depths of the TDR profiles within the same cluster (i.e., the same topographic unit, namely Northern slope, valley bottom and Southern slope) are calculated based on a linear interpolation and used for calculating the temperature correction of the TDR measured soil water content values from the TDR traces according to Kaatze (1989). By doing this, we assume that i) the soil temperature changes linearly with depth between the observations at 5, 10, 20, 30, 50 and 100 cm, regardless of material properties changes in-between, and ii) the soil temperature measured at each of the three plots (i.e., P2, P4 and P7) is representative for the cluster (i.e., cluster North consisting of P1, P2 and P3, measured at P2; cluster Valley consisting of P4 and P5, measured at P4; cluster South consisting of P6, P7 and P8, measured at P7).

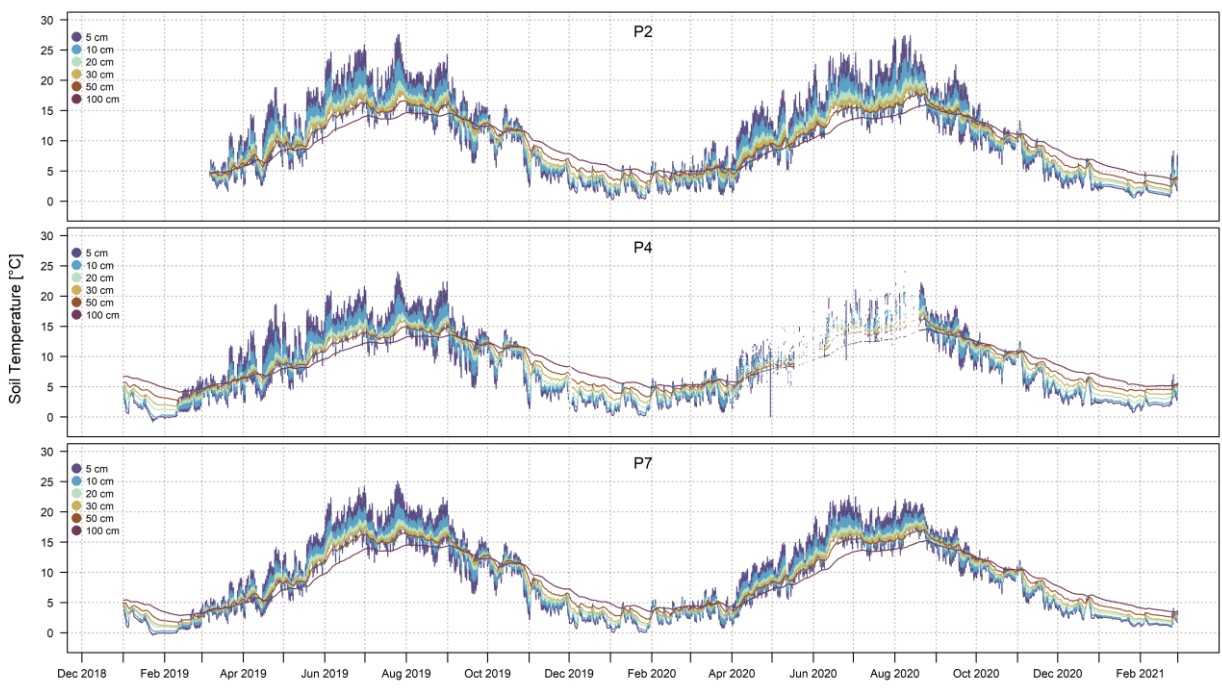

**Figure 5: Time series of soil temperature data. The plots were made using the data set as it appears in the online archive. The data are plotted using a scientific colour scale from Crameri (2018) chosen according to the principles described in Crameri et al. (2020).**

## 3.3 Water level

Every station of the *STH-net* is equipped with a monitoring well consisting of a LDPE (i.e, low-density polyethylene) tube drilled to the maximum depth of 2 m and instrumented with levelogger LTC (Solinst, Ontario, Canada), model 3001- M10. Due to an initial malfunctioning of the sensors, only the data measured since March 9[th], 2020 are available. In contrast to the other measurements of the data set presented here, the water level data are downloaded manually. Figure 6 shows the time series of the water level data and reports the maximum depth for every well. Seasonal dynamics of the groundwater level are evident for the wells in the valley bottom (P4 and P5) and for P6, located next to the creek. The wells on the slopes (P1, P2, P3, P7 and P8) stay dry for most of the monitored period and only show quick rises and recessions of the water level in the winter and spring season.

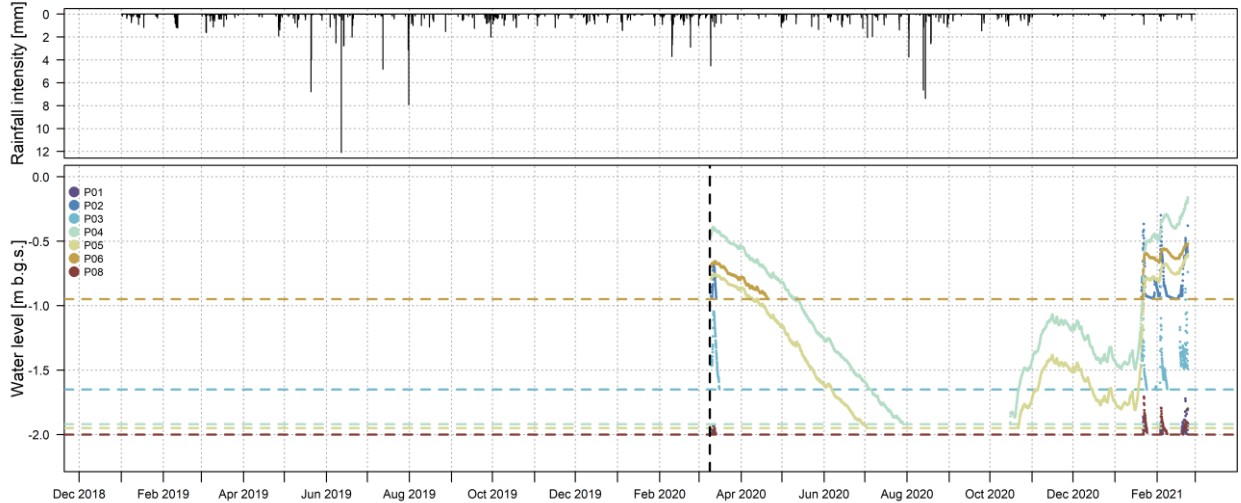

**Figure 6: Time series of water level data. The plots were made using the data set as it appears in the online archive. The dashed vertical line indicates the start of the measurements (March 9[th], 2020). The dashed horizontal lines indicate the depth of the wells. The data are plotted using a scientific colour scale from Crameri (2018) chosen according to the principles described in Crameri et al. (2020).**

## 3.4 Meteorological data

In the central part of the Schäfertal Hillslope site (Fig. 1), a WXT 520 weather station (Vaisala Oyj, Laskutus, Finland) equipped with a CMP3-L pyranometer (Kipp & Zonen, Delft, Netherlands) installed at the height of 2 m measures the wind vector, air temperature and pressure, relative humidity, liquid precipitation, hail and solar radiation. The system is fully integrated with the data logger of the central monitoring station and the meteorological variables are measured at the same times as the TDR and soil temperature profile probes. Fig. 7 shows the hourly time series of the meteorological variables.

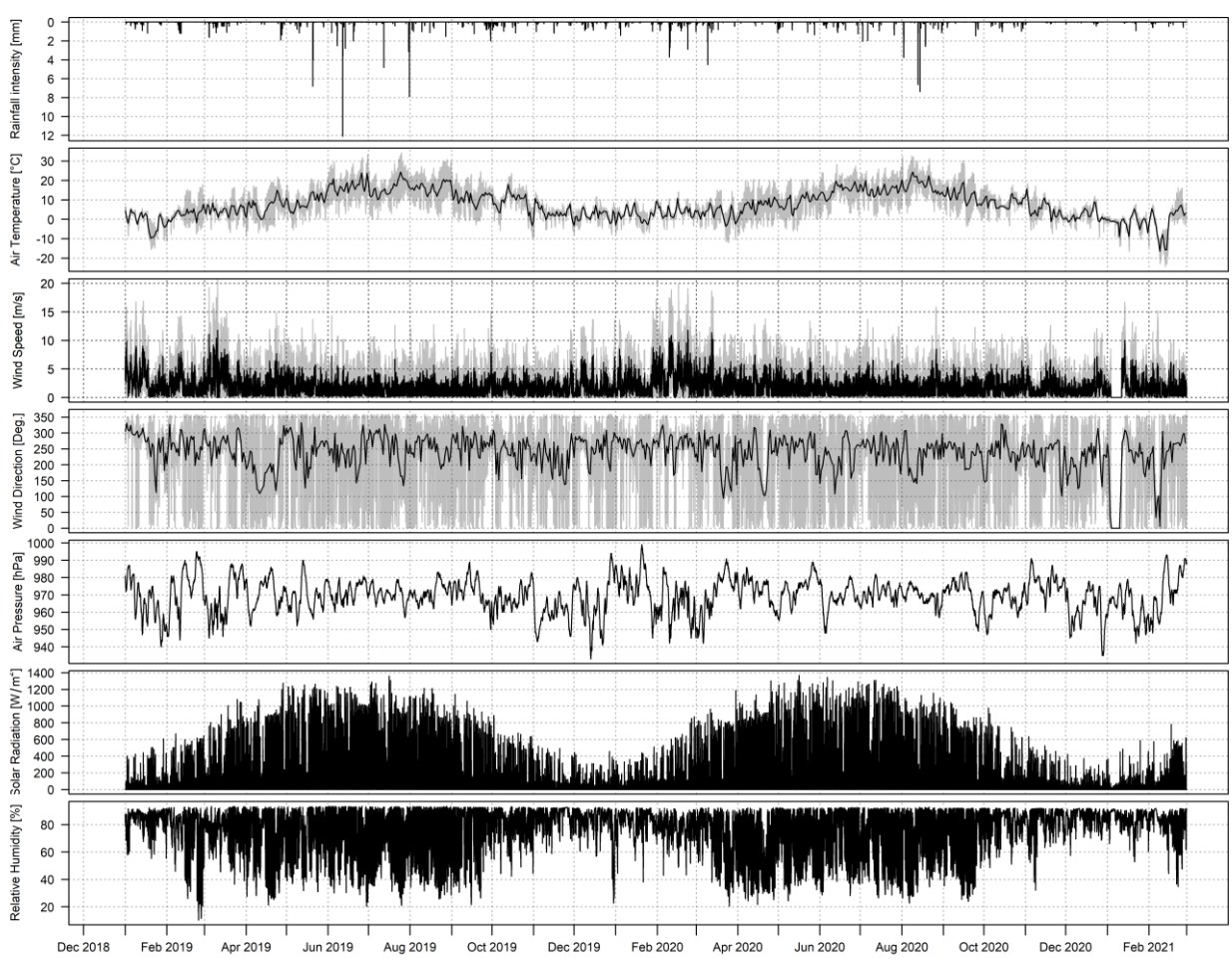


**Figure 7: Time series of all the meteorological variables measured at the *Schäfertal Hillslope* site. The plots were made using the data set as it appears in the online archive. The black line in the second, third and fourth plots shows the daily average temperature, the average wind speed and the daily average wind direction, respectively while all other data are in 10-min time steps.**

**3.5 Soil properties**
During the installation of the STH-net, one bulk soil sample and one volumetric soil sample were collected at every soil pit at
the same depth as each of the TDR probes were installed. From the bulk samples, the percentage of sand, silt and clay in the
fine earth fraction was determined in the laboratory using the pipette method. The volumetric soil samples were collected with
a stainless stain ring and used for the soil porosity and bulk density estimation. Fig. 8 shows the classification of the soil
samples according to the German soil textural classes (Ad-hoc-AG Boden, 2005), considered suitable for the soil
parameterization for physically-based hydrological modelling (Bormann, 2007).

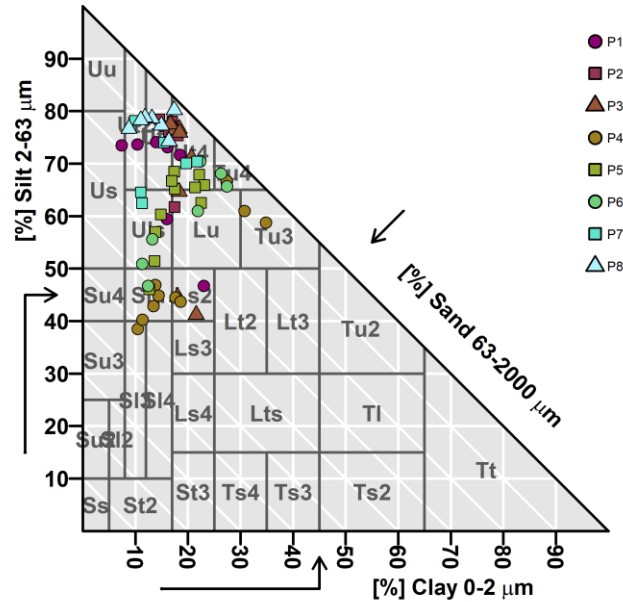

**Figure 8: Soil textural classification according to the German *Bodenkundliche Kartieranleitung* (Ad-hoc-AG Boden, 2005) grouped by soil profiles (P1 to P8).** *Ss*: pure sand; *Su2*: slightly silty sand; *Sl2*: slightly loamy sand; *Sl3*: medium loamy sand; *St2*: slightly clayey sand; *Su3*: medium silty sand; *Su4*: highly silty sand; *Slu*: loamy silty sand; *Sl4*: highly loamy sand; *St3*: medium clayey sand; *Ls2*: slightly sandy loam *Ls3*: medium sandy loam; *Ls4*: highly sandy loam; *Lt2*: slightly clayey loam; *Lts*: clayey sandy loam; *Ts4*: highly sandy clay; *Ts3*: medium sandy clay; *Uu*: pure silt; *Us*: sandy silt; *Ut2*: slightly clayey silt; *Ut3*: medium clayey silt; *Uls*: loamy sandy silt; *Ut4*: highly clayey silt; *Lu*: silty loam; *Lt3*: medium clayey loam; *Tu3*: medium silty clay; *Ts2*: slightly sandy clay; *Tu4*: highly silty clay; *Tu2*: slightly silty clay; *Tl*: loamy clay; *Tt*: pure clay. The figure was created in RStudio with the package "The Soil Texture Wizard" (https://CRAN.R-project.org/package=soiltexture) by Julien Moeys. The data are plotted using a scientific colour scale from Crameri (2018) chosen according to the principles described in Crameri et al. (2020).

**4 Uncertainties and data usability**

For the estimation of soil water content using a composite dielectric approach, some physical parameters must be known. These are primarily temperature, porosity and the dielectric number of the solid matrix ($\varepsilon_{soil}$). Among them, soil temperature plays the major role in determining the global uncertainty. As part of the *STH-net*, soil temperature is measured in situ at the same time as the TDR waveforms, which enables an accurate temperature correction. The soil porosity was estimated for every sampling point from undisturbed soil cores and introduces an uncertainty. For $\varepsilon_{soil}$ we have chosen the value of 4.6, corresponding to the dielectric permittivity of quartz. This value was chosen arbitrarily hence introduces an uncertainty. For a more extensive discussion about the uncertainty of the soil water content estimation as due to the single parameters we refer to Roth et al. (1990). For the data set presented here, we estimated the uncertainty of the calculated soil water content using the CRIM formula by varying the values of $\varepsilon_{soil}$ and porosity between 4 and 6 and between 0.3 and 0.5, respectively (similar

to Wollschläger et al., 2010). We obtained values < ±0.03 m³/m³ as largest uncertainty of the soil water content estimation.
This information is reported in Table 1 along with the measurement range, accuracy and resolution of the other variables
provided within the data set described in this article.
Rain gauges may misestimate the rainfall rate under certain circumstances, especially when rainfall events are associated to
strong wind. The experiment described in Basara et al. (2009) shows that a sensor similar to the one installed at the *Schäfertal*
*Hillslope* site overestimates the rainfall intensity in an urban environment. The rainfall rate data presented in this article were
compared to those of several other rain gauges (data from partner research institute, not available here) located ca. 100 m away
from the site. The rainfall intensity values measured by our sensor do not underestimate the rainfall rate values nor completely
miss rainfall events. With our data set, we make the measured data available to any interested scientists along with all relevant
site information and let them the choice about eventual compensation measures to be applied. The correction function proposed
by Richter (1995) is commonly used for studies conducted in Central Germany to account for the possible wind-induced
underestimation of the rainfall intensity.
Until a few years ago, the Schäfertal catchment used to be affected by significant snowfall, with major snowmelt events
occurring between January and April, whose effects on the hydrological processes are described, e.g., in Ollesch et al. (2005).
In the last years, however, no significant snowfall events were observed. The last winter period (December 2020 to February
2021), instead, was characterized by exceptionally intense snowfall (with a maximum of ca. 45 cm on February 8[th], 2021) that
accumulated and persisted. Unfortunately, the technical infrastructure currently available at the site does not allow a
meaningful estimation of the snow height and distribution during the monitoring period, hence the snowfall events are not
recorded by the weather station in use (see Fig. 7). Because of this, the snow contribution to the water balance needs to be
derived from the meteorological and soil temperature data available.
Overall, 9.3 % of the soil water content data and 7.6 % of the soil temperature data are missing (particularly until March 2019
for the station North and between April and August 2020 for the station Valley) due to various technical failures.
**Table 1: Measurement range, accuracy and resolution of the measurement devices described in Section 3.**

| | Measurement range | Accuracy | Resolution |
|---|---|---|---|
| STH-net station | | | |
| Soil water content[1] | 0 to 1 m³/m³ | < ±0.03 m³/m³ | - |
| Soil temperature[2] | -20°C to +50°C | ± 0,1°C | 0,034°C |
| Water level[3] | 0 to 50°C (Barologger 5: -10 to +50°C), FS = 10 m | ± 0.5 cm | 0.0006% FS |
| Weather station | | | |
| Barometric Pressure[4] | 600 to 1100 hPa | ±0.5 hPa at 0 to +30 °C ±1 hPa at -52 to +60 °C | 0.1 hPa, 10 Pa, 0.001 bar, 0.1 mmHg, 0.01 inHg |
| Air Temperature[4] | -52 to +60 °C | ±0.3 °C | 0.1 °C |
| Wind speed[4] | 0 to 60 m/s | ±3 % at 10 m/s | 0.1 m/s |
| Wind direction[4] | 0 to 360° azimuth | ±3.0° | 1° |
| Relative Humidity[4] | 0 to 100 % RH | ±3 %RH at 0 to 90 %RH | 0.1 %RH |

| | | ±5 %RH at 90 to 100 %RH | |
|---|---|---|---|
| Rainfall intensity[4] | 0 to 200 mm/h (broader range with reduced accuracy) | Daily accumulation: better than 5 %, weather dependent | 0.01 mm |
| Hail[4] | n.a. | n.a | 0.1 hit/$cm^2$ |
| Solar radiation[5] | Maximum solar irradiance: 2000 W/m² | ±5 % | < ±5 W/m² |

[1] custom-made TDR probes (Helmholtz Centre for Environmental Research GmbH – UFZ, Leipzig, Germany)
[2] Th3-s soil temperature profile probe (formerly UMS GmbH, Munich, Germany). Source:
https://www.google.com/url?sa=t&rct=j&q=&esrc=s&source=web&cd=&ved=2ahUKEwjQjpTu4bvuAhWm4YUKHTKhCsUQF
jABegQIARAC&url=http%3A%2F%2Fcnyhome.cafe24.com%2Fpdffile%2FTh3sManual.pdf&usg=AOvVaw1JN8EI6XoJ6F3Ly
Jw9PnnK (accessed Apr 13th, 2021).
[3] 3001-M10 levelogger LTC (Solinst, Ontario, Canada). Source: https://www.solinst.com/products/data/3001-ltc.pdf (accessed
Apr 13th, 2021).
[4] WXT 520 weather station (Vaisala Oyj, Laskutus, Finland). Source:
https://www.vaisala.com/en/file/9411/download?token=DOb1ETJK (accessed Apr 13th, 2021).
[5] CMP3-L pyranometer (Kipp & Zonen, Delft, Netherlands). Source: https://www.kippzonen.com/Product/11/CMP3-
Pyranometer (accessed Apr 13th, 2021).

**5 Data management**

The *STH-net* data stored by the three data loggers are accessed and downloaded remotely using the software *Loggernet* (Campbell Scientific Inc., Logan, UT, United States). The only exception are the water level data, which are manually downloaded. The data files are regularly quality checked and uploaded to the EUDAT record *STH-net* (https://b2share.eudat.eu/records/82818db7be054f5eb921d386a0bcaa74), where they remain available for download.

**6 Data sets**

The *STH-net* data are archived as separate text files for the different data types: soil water content, soil temperature, water level and meteorological variables. Furthermore, the geographic coordinates of the measurement locations and the soil information are available for download. The time series data start from January 1st, 2019 and continue with hourly time steps until the most recent update. At the time of the manuscript submission, the latest entry refers toFebruary 28th, 2021. The water level data are available with a 2-hours resolution and covers the time period between March 6th, 2020 and February, 23rd, 2021. All the data published in the online archive (DOI 10.23728/b2share.82818db7be054f5eb921d386a0bcaa74) will be updated approximately on a 3-months basis.

## 7 Data availability

The *STH-net* data are available under a dynamic identifier DOI 10.23728/b2share.82818db7be054f5eb921d386a0bcaa74 (Martini et al., 2020) at the time of the manuscript submission (from there, all future versions of the archive can be easily accessed) under the Creative Commons Attribution license (CC-BY 4.0).

## Author contribution

Edoardo Martini: conceptualization, data curation, formal analysis, funding acquisition, investigation, methodology, visualization, writing – original draft preparation, writing – review & editing.

Matteo Bauckholt: data curation.

Simon Kögler: conceptualization, data curation, methodology.

Manuel Kreck: data curation, methodology, writing – review & editing.

Kurt Roth: conceptualization, resources, writing – review & editing

Ulrike Werban: conceptualization, funding acquisition, resources, writing – review & editing

Ute Wollschläger: conceptualization, writing – review & editing

Steffen Zacharias: conceptualization, funding acquisition, resources, writing – review & editing

## Competing interests

The authors declare that they have no conflict of interest.

## Acknowledgments

The installation and operation of Schäfertal Hillslope site research infrastructure was funded and supported by the Terrestrial Environmental Observatories (TERENO), which is a joint collaboration program involving several Helmholtz Research Centres in Germany, and the TERENO observatory Harz/Central German Lowland operated by the UFZ Helmholtz Centre for Environmental Research is gratefully acknowledged for providing access to the *Schäfertal Hillslope* site and to the *STH-net* data. Edoardo Martini received funding from the German Research Foundation (DFG) through the research grant MA 7936/1-1.

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
