# Peer review of "STH-net: a soil monitoring network for process-based hydrological modelling from the pedon to the hillslope scale"

_Earth System Science Data, 2020_

## Author Response (AR1)

We gratefully acknowledge Dr. Van Looy and the six anonymous referees for providing valuable comments to our manuscript. As anticipated in our short response, we accounted for all of those comments for improving the quality of the manuscript.

In the following pages we provide a point-by-point response to the 7 reviews. At the end of the document, we include a marked-up revised manuscript version. We replaced the figures (all of them except for Fig. 2) with the new ones and added one new figure (now named "Figure 6") and one table. We added a new section "Uncertainties and data usability". We added one co-author who contributed significantly to the production of the data set. We modified the title of the manuscript.

The revised manuscript refers to the latest version of the data set published on Apr 13th, 2021 (DOI http://doi.org/10.23728/b2share.82818db7be054f5eb921d386a0bcaa74) and available at https://b2share.eudat.eu/records/82818db7be054f5eb921d386a0bcaa74.

Edoardo Martini, on behalf of the co-authors.

**Authors' response to community comment CC1 "Referee Comment on essd-2020-363" by Kris Van Looy (15 Jan 2021)**

**CC1:**
This preprint describes in a well-structured manner the development and access to a rich dataset of a critical zone observatory on a Central German hillslope that was instrumented with sets of TDR probes and coupled temperature probes and piezometers. As the authors state such dataset offers broad opportunities for modelling applications for hydrological models and land surface models. The detail and information in the dataset is well illustrated with some graphs showing hourly soil water content and soil temperature data at different depths for different plots through the two-year dataset.

Some specific comments:
The title mentions model application limited to only hydrological models; whereas to my opinion also land surface models and earth system modelling applications, oriented at energy exchanges might benefit from these detailed data on both hydrological and temperature/meteorological data at this high resolution.

**AC:**

We thank Dr. Van Looy for this useful hint. We agree that the data set can enable analysis of processes beyond soil hydrology, although the monitoring network was designed specifically for that scope.

I would also like to see the other measured parameters (and references to data sets) at this site; it is mentioned that it is operational since 2010 and that other parameters were measured. Linkage to these measurements would strongly increase the attractiveness and broader applicability of the dataset.

The measurements running since 2010 at the Schäfertal catchment, as part of the TERENO project, are described in the work by Wollschläger et al. (2017) cited in the text. As mentioned in that publication, those data are quality checked and to be made available in the TEODOOR data portal (https://ddp.tereno.net/ddp) as for the other data from the TERENO project. The measurements in the catchment are run by various groups and it is in their responsibility to publish the data of their work. However, with this manuscript we aim at informing the community about the existence of the recently implemented monitoring network *STH-net* (which is our current research work) and about the availability of its data.

Line 22 says high-quality, but for now I would restrict that to mentioning high-resolution data, since the piezometers up to date were not yet high quality, and in figure 5 the results of Probe 4 are incomplete as well.

Thanks for pointing this out, as other reviewers did. We changed the text accordingly.

The *STH-net* is intended to provide scientists with  data needed for developing and testing modelling approaches in the context of vadose-zone hydrology at spatial scales ranging from the pedon to the hillslope.

It should also be mentioned in the paper from which date on the data acquisition was properly running and till when these measurements will be sustained, and published (at what interval).

> In the manuscript we state "The installation was carried out between June and August 2018 and all the measurements collected until the end of December 2018 were discarded to allow the soil to re-compact naturally during the first rainy season." (Lines 137-139).
>
> There is no plan about the end of the monitoring. The expected lifetime of the monitoring network is in the order of decades.
>
> We just published an updated version of the data set. From now on, data updates will follow approximately on a 3-months basis. We added this information to the manuscript.
>
> At the time of the manuscript submission, the latest entry refers to February 28th, 2021. The water level data are available with a 2-hours resolution and covers the time period between March 6th, 2020 and February, 23rd, 2021. All the data published in the online archive (DOI http://doi.org/10.23728/b2share.82818db7be054f5eb921d386a0bcaa74) will be updated approximately on a 3-months basis.

Line 62-63 states: 'environmental research platforms focusing on the interconnection between physical, chemical and biological processes affecting Earth surface, offer the opportunity to integrate information about different compartments of the environment, scales, and knowledge from different disciplines 'but the reference just refers to hydrological models, so suggest to refer to the research you also contributed to, in Baatz et al 2018: Steering operational synergies in terrestrial observation networks: opportunity for advancing Earth system dynamics modelling, Earth System Dynamics, 9.

> We have deleted the entire sentence about CZOs (and references therein) from the revised manuscript as it was somewhat out of the focus of the overall introduction section.

Line 67 talks of 'matter fluxes 'but for the measurements this is abandoned; in this perspective the measurement of EC should also be done and documented since it can interfere with the TDR measurements, and since it can highlight significant temporal process changes.

> As mentioned in the text (Line 136), we also estimate the soil EC from the TDR traces. We are currently working on quality-checking the EC data and we plan to make them available in the data archive when possible, as we are aware that those data would add value to the dataset. In any case, the measured soil EC at the *Schäfertal Hillslope* site are very low as shown from the electromagnetic induction measurements published by Martini et al., 2017. Based on a two-years data set of available TDR waveforms for both, *in situ* soil water content and EC measurements, we can exclude disturbances to the soil water content measurements.

Line 68 says "soil mapping" but in the soil description only texture is given; this is too often a significant problem with site data descriptions. Soil is more than just texture; we need profile descriptions and soil type identification to interpret the observed data and to 'relate 'it to other observations. The general description of soil types in the area in Line 88 is not specified to the sampling site.

> We thank Dr. Van Looy for pointing this out, we are well aware that soils are more than a pattern of textures. A more detailed description of the soil types appears in the other publications about the *Schäfertal Hillslope* site (referenced in the manuscript). In this case, we targeted the soil description

to hydrologically relevant layers as required for simulation models hence reported the new data about grain size distribution, bulk density and porosity values corresponding to each of the 65 TDR probes installed, and grouped the sampling points (i.e., soil profiles and depths) into soil horizons as identified in the field during the soil description. This will allow other scientists to use, e.g., pedotransfer functions or other parameterization approaches. We believe that this is a solid ground for representing the hydraulically relevant layers in hydrological models. A more extensive soil description is provided in Martini et al. (2015), cited in the manuscript. We revised the text accordingly.

The slopes are covered by silty loam Cambisols more evolved towards the footslope, while loam and silty loam stagnic Gleysols occupy the valley bottom. An extensive description of the soil units mapped at the site is provided in Martini et al. (2015).

**Authors' response to the referee comments RC1: "Comment on essd-2020-363" by Anonymous Referee #1 (19 Jan 2021)**

**RC1:**

The manuscript presents the data set of a new soil monitoring network, providing soil water content measurements from TDR probes, soil temperatures, and meteorological forcing. The paper is well written and the provided data set is of high relevance for hydrology.

I only have few minor comments that need further clarification or additions to the data set.

**AC:**

We thank the referee for this positive comment.

Specific comments:

Measurement interval vs. reported data interval: The measurements are taken in a 10 minute interval. However, the data is reported in an hourly resolution. What is the motivation for this?

Please provide more details on the averaging. Line 197 only states "The original files are averaged to hourly values […]". For cumulative data like the precipitation I assume it is not the average, but the sum of the the values in the previous hour. For the other values, is it the average of values half an hour before and half an hour after the reported time? Please clarify.

Also, the averaging interval of 12 hours to smooth the water content data seems quite long. This could impact the accuracy of the timing of e.g. the arrival of an infiltration front at a sensor location and reduce the information contained in the data set. For example, on October 4/5, 2019 the water content for T08 increases by about 0.1 in 10 hours (in the reported smoothed data). This indicates that the 12h smoothing interval may have a significant impact on some situations with rapid changes in the water content. Please briefly discuss the impact of the chosen smoothing, consider shorter smoothing intervals or possibly even provide an additional water content data file without this smoothing.

According to the comments of some of the other reviewers, in the revised manuscript we now report and describe the data in the 10-minutes interval and not the hourly aggregates, anymore.

Temperature correction of water content measurements: The relative dielectric permittivity of water depends on the temperature. Temperature measurements are available. However, the manuscript does not mention if the dielectric permittivity for water is calculated temperature dependent. If so, please mention this and provide the details, including information on how the more sparse temperature measurements are inter- and extrapolated to the locations of the TDR probes.

Also in this case the referee raises a good point. We admit that we missed to provide this important information in the text. The temperature dependence of dielectric permittivity of water is accounted for in the calculations. We improved the revised manuscript accordingly.

For every temperature profile, the soil temperature values corresponding to the depths of the TDR profiles within the same cluster (i.e., the same topographic unit, namely Northern slope, valley bottom and Southern slope) are calculated based on a linear interpolation and used for calculating the temperature correction of the TDR measured soil water content values from the TDR traces according to Kaatze (1989). By doing this, we assume that i) the soil temperature changes linearly with depth between the observations at 5, 10, 20, 30, 50 and 100 cm, regardless of material properties changes in-between, and ii) the soil temperature measured at each of the three plots (i.e., P2, P4 and P7) is representative for the cluster (i.e., cluster North consisting of P1, P2 and P3, measured at P2; cluster Valley consisting of P4 and P5, measured at P4; cluster South consisting of P6, P7 and P8, measured at P7).

There is very little information on the piezometer (e.g. manufacturer and model). The data itself is missing in the data set, although available since March 2020. Is it the goal to include the data into the data set? If so, I would recommend to add the data as part of this revision.

We added the water level data to the data set and to the revised manuscript.

Overall, there is limited information about the different probes and sensors. I would recommend to add a table, that lists measurement range, uncertainty, and resolution for the different sensors.

We added a table providing information on the used sensor and probes to the revised manuscript.

Is the time reported in UTC or local time? Please mention this.

All the times are in UTC. We added this information in the metadata of the online archive. We thank the reviewer for this observation.

Line 144: "$\varepsilon_{soil}$ is set to 4.6." Please motivate why this value was chosen.

The value of $\varepsilon_{soil}$ = 4.6 corresponds to the dielectric permittivity of quartz and it is often used for calculations of the mineral fraction of the soil matrix in the CRIM formula. This is feasible as the dielectric permittivity of the most minerals is in this range and very small compared to the dielectric permittivity of water. Hence, its influence on the estimated water contents is negligible (see also Fig. 5 in Wollschläger et al., 2010). We added more information in the new section 4 "Data usability".

4 Uncertainties and data usability
For the estimation of soil water content using a composite dielectric approach, some physical parameters must be known. These are primarily temperature, porosity and the dielectric number of the solid matrix ($\varepsilon_{soil}$). Among them, soil temperature plays the major role in determining the global uncertainty. As part of the STH-net, soil temperature is measured in situ at the same time as the TDR waveforms, which enables an accurate temperature correction. The soil porosity was estimated for every sampling point from undisturbed soil cores and introduces an uncertainty. For $\varepsilon_{soil}$ we have chosen the value of 4.6, corresponding to the dielectric permittivity of quartz. This value was chosen arbitrarily hence introduces an uncertainty. For a more extensive discussion about the uncertainty of the soil water content estimation as due to the single parameters we refer to Roth et al. (1990). For the data set presented here, we estimated the uncertainty of the calculated soil water content using the CRIM formula by varying the values of $\varepsilon_{soil}$ and porosity between 4 and 6 and between 0.3 and 0.5, respectively (similar to Wollschläger et al., 2010). We obtained values $< \pm 0.03 \ \mathrm{m^3/m^3}$ as largest uncertainty of the soil water content estimation. This information is reported in Table 1 along with the

measurement range, accuracy and resolution of the other variables provided within the data set described in this article.

Technical comments:

Line 67-70: "Specifically, the approach followed at the site accounts for the soil spatial variability through detailed soil mapping and is designed to provide in situ data of, to our knowledge, the best quality available to date, with high temporal resolution and dense coverage in the vertical direction, about the soil water dynamics in the vadose zone and of its boundary conditions."

There is no further information about the soil mapping in the manuscript. Is this part of a previous publication for this site? If so, please give a reference here.

A more extensive soil description is provided in Martini et al. (2015), cited in the manuscript. We revised the text accordingly.

The slopes are covered by silty loam Cambisols more evolved towards the footslope, while loam and silty loam stagnic Gleysols occupy the valley bottom. An extensive description of the soil units mapped at the site is provided in Martini et al. (2015).

The stations P1, P2 and P3 are located within the Northern (i.e., South-facing) slope and cover the transition between the soil units STU1 and STU2 described in Martini et al. (2015); the stations P4 and P5 fall into the valley bottom, i.e., soil unit STU3; P6, P7 and P8 cover the lower part of the Southern (i.e., North-facing) slope, i.e., soil unit STU4.

In line 83 "Wollschläger et al., 2018" should be "Wollschläger et al., 2017" (at least according to the References).

Correct, it should be 2017. We corrected this and checked all the other citations.

Lines 112-114: "Each of the instrumented soil profiles located on the hillslopes features seven TDR probes installed at the depths of 0.1, 0.2, 0.3, 0.4, 0.5, 0.6 and 0.7 m, whilst the profiles at P4 and P5 feature additional TDR probes at the depths of 0.8, 0.9, 1 and 1.1 m in order to cover the deeper soils."

P3 seems to also have a probe in the depth of 0.8 m.

Well spotted, thanks for pointing this out. We corrected the text in the revised manuscript.

Each of the instrumented soil profiles located on the hillslopes features a minium of seven TDR probes installed at the depths of 0.1, 0.2, 0.3, 0.4, 0.5, 0.6 and 0.7 m, whilst an additional probe is installed at P3 at the depth of 0.8 m and the profiles at P4 and P5 feature additional TDR probes at the depths of 0.8, 0.9, 1.0 and 1.1 m in order to cover the deeper soils.

Some of the figures have a rather low quality/resolution. This should be improved.

We changed to higher-resolution images in the revised manuscript.

**Authors' response to the referee comments RC2: "'Comment on essd-2020-363" by Anonymous Referee #2 (29 Jan 2021)**

**RC2:**

Comment on "STH-net: a model-driven soil monitoring network for process-based hydrological modelling from the pedon to the hillslope scale" by Martini et al.

The paper is generally well written and presented and describes a dataset will be useful for many wishing to understand soil processes better. A have some general comments and specific points which I have made below.

**AC:**

We thank the referee for this positive comment.

General comments

I am not sure the title for the paper is suitable. The title says the monitoring network is model driven, to me this suggests you used a model to find areas of highest uncertainty and instrumented these. How about shortening to "STH-net: a soil monitoring network for process-based hydrological modelling from the pedon to the hillslope scale"

We thank the reviewer for this suggestion. We changed the title accordingly.

*STH-net*: a  soil monitoring network for process-based hydrological modelling from the pedon to the hillslope scale

Seems to be a lot made about the high resolution data collected at 10 minute intervals, yet the data set is averaged to hourly – not sure why this is. A user can average up if they need but they can't disaggregate once you present averaged data. 12 hour smoothing and hourly averaging may remove a lot of important information, particularly around rainfall events when responses are quick.

According to the comments of some of the other reviewers, in the revised manuscript we report the data in the 10-minutes interval and not the hourly aggregates, anymore.

The section on groundwater level measurement is very sparse. There are no details of the type of sensor used and no corresponding figure like the other sensors types get. This dataset also appears to be missing on the data sharing site. Details need to be improved and data added or any mention of the piezometers should be removed.

We added the water level data to the data set and to the revised manuscript and included the missing information.

3.3  Water level
Every station of the *STH-net* is equipped with a monitoring well consisting of a LDPE (i.e, low-density polyethylene) tube drilled to the maximum depth of 2 m and instrumented with levelogger LTC (Solinst, Ontario, Canada), model 3001-M10. Due to an initial malfunctioning of the sensors, only the data measured since March 9th, 2020 are available

. In contrast to the other measurements of the data set presented here, the water level data are downloaded manually. Figure 6 shows the time series of the water level data and reports the maximum depth for every well. Seasonal dynamics of the groundwater level are evident for the wells in the valley bottom (P4 and P5) and for P6, located next to the creek. The wells on the slopes (P1, P2, P3, P7 and P8) stay dry for most of the monitored period and only show quick rises and recessions of the water level in the winter and spring season.

Has the accuracy of the WXT520 rainfall been assessed in any way? My experience in using such sensors shows that they work well in some climate conditions and not in others. Seeing as the rain is the primary driver of any modelling it is important to establish this as an accurate measure. These devices have been tested by others and show not produce some serious over-estimates (see Basara et al 2009)

The experiment described in Basara et al. (2009) shows that the WXT 510 sensor overestimates the rainfall intensity in an urban environment. From our experience, based on the comparison with other rain gauges (data from partner research institute, not  shown or used for the study) located ca. 100 m away from the *Schäfertal Hillslope*, the rainfall intensity values measured by our WXT 520 sensor do not underestimate the "real" values. Actually, the correction function proposed by Richter (1995), unfortunately only available in German, is commonly used for studies conducted in Central Germany to account for the possible wind-induced underestimation of the rainfall intensity. We added this information in the new section 4 "Uncertainties and data usability".

Richter, D.: Ergebnisse methodischer Untersuchungen zur Korrektur des systematischen Meßfehlers des Hellmann-Niederschlagsmessers, Berichte des Deutschen Wetterdienstes 194, Deutscher Wetterdienst, Offenbach am Main, ISBN: 3881483098, 1995 (in German). http://nbn-resolving.de/urn:nbn:de:101:1-201601274368

Rain gauges may misestimate the rainfall rate under certain circumstances, especially when rainfall events are associated to strong wind. The experiment described in Basara et al. (2009) shows that a sensor similar to the one installed at the *Schäfertal Hillslope* site overestimates the rainfall intensity in an urban environment. The rainfall rate data presented in this article were compared to those of several other rain gauges (data from partner research institute, not available here) located ca. 100 m away from the site. The rainfall intensity values measured by our sensor do not underestimate the rainfall rate values nor completely miss rainfall events. With our data set, we make the measured data available to any interested scientists along with all relevant site information and let them the choice about eventual compensation measures to be applied. The correction function proposed by Richter (1995) is commonly used for studies conducted in Central Germany to account for the possible wind-induced underestimation of the rainfall intensity.

Specific comments

L13 – change to "…dynamics are being…"

Corrected in the revised manuscript, thank you.

The *Schäfertal hillslope* site is part of the TERENO Harz/Central German Lowland Observatory and its soil water dynamics  are being monitored intensively as part of an integrated, long-term, multiscale and multi-temporal research framework linking hydrological, pedological, atmospheric and biodiversity-related research to investigate the influences of climate and land use change on the terrestrial system.

**L29-31 – sentence needs to be reworded or split in two so that it makes sense**

We rephrased the sentence in the revised manuscript.

Soils provide crucial ecosystem functions such as water storage and filtering, food and other biomass production, recycling of carbon and nutrients, biological habitat and gene pool,  physical and cultural heritage, source of raw materials and platforms for human life (United Nations, 2014; Vereecken et al., 2016).

**L35-36 – reword to "hence numerical models are needed for the comprehensive representation of the system state and fluxes so that the hydrological system can be better understood." ???**

Reworded in the revised manuscript, thanks for this suggestion.

In order to address effectively this complexity, state-of-the-art experimental approaches must be coupled to numerical models  for the comprehensive representation of the system properties, states and  fluxes  so that the hydrological system can be better understood.

**L38 – change remarked to noted**

Corrected in the revised manuscript, thank you.

Recently, Vogel (2019) provided a comprehensive discussion about the scales and scaling issues in the context of soil hydrological research and  noted the need for looking at small-scale soil properties (i.e., at the pedon scale, at which soil physics is capable of describing states and fluxes with sufficient accuracy) as a necessary step towards understanding and summarizing the processes at larger scales.

**L41 – makes no sense. Delete 'however'?**

We have changed the sentence in the revised manuscript, also according to other comments.

However, the author remarks that  high-resolution measurements of the relevant states and properties cannot be achieved at the larger scale (i.e., catchment, the typical scale of application of hydrological research).

**L56 – change remarked to noted**

Changed to "highlighted" in the revised manuscript, thank you.

Bronstert (1999)  highlighted the importance of linking experimental knowledge to the experience gained from numerical modelling applications as a very valuable synergistic combination.

**L71 – change to "...aim to provide physical models…"**

Changed in the revised manuscript.

With this design tailored to the needs of vadose zone modelling, we aim  to provide physical models with ideally all the data needed for quantifying and predicting the soil water fluxes at spatial scales ranging from the pedon to the hillslope scale, with important implications, in terms of methodological advance and process understanding, for catchment-scale processes.

**L131 – delete safety**

Corrected in the revised manuscript.

The clusters are powered by  extra low voltage cables buried ca. 0.3 m below the ground and cased in HDPE (i.e., high-density polyethylene) tubes and an AGM (i.e., absorbent glass mat) battery capable of supplying the required power in case of power cut-off.

**L134- 135 – suggest change to "The TDR probes were custom made and have three 0.2 m-long rods. They were calibrated through measurements in air and in water…"**

Corrected in the revised manuscript, thank you.

The TDR probes were custom made and have three 0.2 m-long rods. They were  calibrated through measurements in air and in water with different salt concentrations for water content and electrical conductivity estimation.

**Authors' response to the referee comments RC3: "'Comment on essd-2020-363" by Anonymous Referee #3 (31 Jan 2021)**

**RC3:**
The submission describes the installation and data collection of a set of eight profiles of TDR soil water content sensors, temperature sensors, and groundwater wells. The installation and instrumentation are well described. The figures do a nice job of providing an overview of the site and the data being collected there.

The manuscript seems to fit many of the criteria listed in the ESSD guidelines, including describing open access data and having a thorough description of the data products. However, the manuscript also falls short in a few areas. The data management section provides an overview of some of the processing applied to the data, but I did not see those scripts listed on the URL provided in the paper. Also, I did not notice any discussion of uncertainties. I think this discussion could be fairly brief, but one example would be to do a sensitivity analysis for Eq. (2), which requires an estimate of porosity and soil dielectric. Given the emphasis of the journal, the authors should work to better quantify uncertainties in the data.

With attention to these details, the manuscript should be acceptable

**AC:**
We now publish and describe the full-resolution data (10-min), only. Hence, no processing was applied. We added a new section 4 "Uncertainties and data usability".

Specific Comments:

L2: The title seems more aspirational than descriptive. I saw nothing related to "model-driven" or "process-based" in the abstract, and while the introduction does a bit better in this regard, the authors might consider editing the title.

In the revised manuscript, we have changed the title to "STH-net: a soil monitoring network for process-based hydrological modelling from the pedon to the hillslope scale" to accommodate the comments from some of the other reviewers.

L13: dynamics "are"… Also this sentence is very long; might consider breaking it up.

Corrected in the revised manuscript, thank you.

The Schäfertal hillslope site is part of the TERENO Harz/Central German Lowland Observatory and its soil water dynamics  are being monitored intensively as part of an integrated, long-term, multi-scale and multi-temporal research framework linking hydrological, pedological, atmospheric and biodiversity-related research to investigate the influences of climate and land use change on the terrestrial system.

L28: This "what" gives rise?

This coupling. Corrected in the revised manuscript.

This coupling gives rise to a suite of physical, chemical, and biological dynamics most of which are highly non-linear and varying in time and space.

**L32: Provide more details or support on what is meant by the strongest gradients are in the vertical direction.**

Clarified in the revised manuscript, also in response to Referee #4.

Soils are widely distributed on the Earth surface Flow and transport processes in unsaturated soils occur predominantly in the vertical direction, with the gravity force playing a major role, as abrupt changes in soil properties due to soil horizons and layers are typically more significant than those in the lateral direction, and because of the strong coupling between soil, vegetation, and atmosphere.

**L34: Could simplify as "heterogeneity of soil properties across spatial scales"**

We revised this sentence.

Here, one aspect that complicates the picture is the heterogeneity of  soil properties.

**L37: Another long sentence that might be better split up.**

We have reworded the sentence, also in response to a comment by Referee #4.

Recently, Vogel (2019) provided a comprehensive discussion about the scales and scaling issues in the context of soil hydrological research and  noted the need for looking at small-scale soil properties (i.e., at the pedon scale, at which soil physics is capable of describing states and fluxes with sufficient accuracy) as a necessary step towards understanding and summarizing the processes at larger scales. In this respect, the author stresses the need for a two-phases approach based on the accurate description of the soil water dynamics at the pedon scale and accounting for the spatial patters of functional soil types that constitute the landscape, including the vertical stratification of soil hydraulic properties and structural attributes. However, the author remarks that  high-resolution measurements of the relevant states and properties cannot be achieved at the larger scale (i.e., catchment, the typical scale of application of hydrological research). In this context, the intermediate scale of hillslopes is crucial for linking the detailed process understanding to larger scale dynamics, recognizing hillslopes as key landscape features that organize water availability on land (Fan et al., 2019).

**L42: Define intermediate and large scales. Which do you consider the hillslope to fall into? You later also use 'hillslope scale' (L58), so try to be consistent with these terms.**

We have reworded the sentence, also in response to a comment by Referee #4. We have introduced the words "catchment" and "hillslope" to clarify intermediate and large scales. The changes are included in the response to the previous comment.

**L56-63: This paragraph was fairly unfocused, and in particularly the part about CZOs seems tangential to the manuscript (e.g., referencing chemical and biological processes).**

We have deleted the entire sentence from the revised manuscript - we agree, it was somewhat out of the focus of the overall introduction section. We would like to keep the first part of the paragraph

because we find it useful for introducing the approach followed at the site, that we describe immediately after.

L69: Awkward sentence.

Reworded in the revised manuscript.

Specifically, the approach followed at the site accounts for the soil spatial variability through detailed soil mapping and is designed to provide *in situ* data  with high temporal resolution and dense coverage in the vertical direction, about the soil water dynamics in the vadose zone and of its boundary conditions.

L85: a subcontinental what?

Sub-continental climate. We have corrected a typo in the text.

The average annual air temperature is 6.9°C, with a sub-continental  climate (Reinstorf, 2010).

L90: either "a meadow" or "meadows"

Changed to "meadows" in the revised manuscript, thank you.

The slopes of the catchment are intensively used for agriculture, whilst meadows  occupy the valley bottom (Schröter et al., 2015).

L102: Elaborate on pedological features.

In the revised manuscript, we have added the reference to Martini et al. (2015) where the pedological features are described.

L105: I did not see a scale on this figure.

We only display the scale in the aerial view and not on the 3-D sketch, where it could look awkward. We believe that this is sufficient.

L124: It would be helpful to label the horizons, if possible.

We have labeled the soil horizons in the new version of the figure

L135: What is meant by self-produced? That the probes produced them?

Changed to "custom made" in the revised manuscript.

The TDR probes were custom made and have three 0.2 m-long rods. They were  calibrated through measurements in air and in water with different salt concentrations for water content and electrical conductivity estimation.

L150: Would be good to briefly comment on the data gaps in P4 and P5 (same for Figure 5).

The same was recommended by several other referees. In the revised manuscript we have added a comment about the gaps in the SWC and soil temperature data.

Overall, 9.3 % of the soil water content data and 7.6 % of the soil temperature data are missing (particularly until March 2019 for the station North and between April and August 2020 for the station Valley) due to various technical failures.

L166: I would use "monitoring well" instead of "piezometers" here and elsewhere, since the two terms are not really synonymous. Also, the sensor used to measure water table height is not described here.

We have changed "piezometers" to "monitoring wells" here and throughout the manuscript.

L179: "were" installed.

Corrected in the revised manuscript, thank you.

During the installation of the STH-net, one bulk soil sample and one volumetric soil sample were collected at every soil pit at the same depth as each of the TDR probes were installed.

**Authors' response to the referee comments RC4: "'Comment on essd-2020-363" by Anonymous Referee #4 (8 Feb 2021)**

**RC4:**

I'm the fifth (!) reviewer of this manuscript and I have very little to add to this manuscript. This is a great site, a very detailed and dataset useful for several hydrological applications. The manuscript is overall well written and illustrated. I only noted some vague terms in Section 1 and I invite the Authors to be more specific in their statements. In addition to the comments by the previous reviewers, I have only few minor comments below.

**AC:**

Thank you for this positive comment. The comments are very helpful and help improving the correctness of the text. We are happy to address them.

Specific Comments:

1. As one of the previous reviewers, I'm not convinced about the "model-driven" part in the title…it looks as it would exclude the need of any purely experimental analysis.

In the revised manuscript, we have changed the title, also to account for the comments from some of the other reviewers.

*STH-net*: a  soil monitoring network for process-based hydrological modelling from the pedon to the hillslope scale

35. I agree but why only models are necessary? On the contrary, I would say that a strong observational basis is necessary in order to understand these non linearities.

We certainly agree. We have reformulated the concept to give more emphasis to the experimental aspect.

Here, one aspect that complicates the picture is the heterogeneity of  soil properties. Another one is the non-linearity of soil processes. In order to address effectively this complexity, state-of-the-art experimental approaches must be coupled to  numerical models  for the comprehensive representation of the system properties, states and  fluxes  so that the hydrological system can be better understood.

73. Yes, true, but it's not described how it is possible to move from the hillslope the catchment scale. Please, elaborate on this.

Thanks for pointing this out. We would like to keep this paragraph as it is but we have now expanded the discussion about the scale issues in soil hydrology earlier in the introduction section.

Recently, Vogel (2019) provided a comprehensive discussion about the scales and scaling issues in the context of soil hydrological research and  noted the need for looking at small-scale soil properties (i.e., at the pedon scale, at which soil physics is capable of describing states and fluxes with

sufficient accuracy) as a necessary step towards understanding and summarizing the processes at larger scales. In this respect, the author stresses the need for a two-steps approach based on the accurate description of the soil water dynamics at the pedon scale and accounting for the spatial patters of functional soil types that constitute the landscape, including the vertical stratification of soil hydraulic properties and structural attributes. However, the author remarks that  high-resolution measurements of the relevant states and properties cannot be achieved at the larger scale (i.e., catchment, the typical scale of application of hydrological research). In this context, the intermediate scale of hillslopes is crucial for linking the detailed process understanding to larger scale dynamics, recognizing hillslopes as key landscape features that organize water availability on land (Fan et al., 2019).

Minor comments and technical corrections

**16. What are the most relevant variables? A bit vague, please specify.**

This is correct. We have changed the figure to a new one that shows all the meteorological variables measured and published in the data set. We have edited the text accordingly.

**22. What does "high quality" mean? Please, specify. Also at 69.**

In response to the comments by other reviewers, we have omitted the expression "high-quality" throughout the manuscript.

**27. "Larger" compared to what? Please, specify.**

We originally used this expression from a pedological point of view, assuming the pedon to be the reference scale. We have now reworded the sentence and omitted "larger".

Soils are embedded in the  environment, coupled to vegetation and atmosphere at the land surface and to groundwater at its lower end.

**32. "Grandient"…in which terms? Soil types? Soil properties? Any reference? This is confusing, please clarify.**

We have now elaborated more on this, also in response to Referee #3.

Soils are widely distributed on the Earth surface. Flow and transport processes in unsaturated soils occur predominantly in the vertical direction, with the gravity force playing a major role, as abrupt changes in soil properties due to soil horizons and layers are typically more significant than those in the lateral direction, and because of the strong coupling between soil, vegetation, and atmosphere. Therefore, despite the relevance of soils for global phenomena, the relevant soil processes are rather local.

**33. Which "phenomena"? Vagueness again, please specify.**

Here we refer to the previous sentence "Soils are widely distributed on the Earth surface" (see changes in response to the previous comment).

**40. "Landscape" is not so appropriate for hydrological applications…what about catchment?**

Landscape is the word used by the author, from the pedological point of view and in the context of soil hydrological research. We find this expression correct in that context and we agree that

"catchment" is the most appropriate one in hydrology. We have reworded the long sentence in agreement to a comment of the Referee #3.

> Recently, Vogel (2019) provided a comprehensive discussion about the scales and scaling issues in the context of soil hydrological research and  noted the need for looking at small-scale soil properties (i.e., at the pedon scale, at which soil physics is capable of describing states and fluxes with sufficient accuracy) as a necessary step towards understanding and summarizing the processes at larger scales. In this respect, the author stresses the need for a two-steps approach based on the accurate description of the soil water dynamics at the pedon scale and accounting for the spatial patters of functional soil types that constitute the landscape, including the vertical stratification of soil hydraulic properties and structural attributes. However, the author remarks that  high-resolution measurements of the relevant states and properties cannot be achieved at the larger scale (i.e., catchment, the typical scale of application of hydrological research). In this context, the intermediate scale of hillslopes is crucial for linking the detailed process understanding to larger scale dynamics, recognizing hillslopes as key landscape features that organize water availability on land (Fan et al., 2019).

**42. What does "intermediate" refer to? Specify. Analogously, what does "larger" mean?**

We have clarified this, also in response to Referee #3 (see changes in response to the previous comment).

**44. Why only "non linearities"? Please explain.**

Not only nonlinear interactions, indeed. However, this sentence  appears indeed to be out of context here and we would leave it out.

**47. Small-scale = plot scale?**

Yes. We introduced this a few paragraphs above "small-scale soil properties (i.e., at the pedon scale, at which soil physics is capable of describing states and fluxes with sufficient accuracy)".

**59-64. Long sentence, please rephrase.**

We have deleted the entire sentence from the revised manuscript as it was somewhat out of the focus of the overall introduction section.

**Fig. 4 and 5. Include precipitation in the top panel. Increase the font size of legend and axis labels.**

Thanks for this suggestion. The new figures look better.

**Fig. 6. "Rainfall" should be "Rainfall rate (mm/hr- or whatever time step it is displays)". Increase the font size of legend and axis labels.**

We have improved the readability of the figures and corrected the axis label. Thank you.

**Authors' response to the referee comments RC5: "'Comment on essd-2020-363" by Anonymous Referee #5 (8 Feb 2021)**

**RC5:**

MS: Data are new (2019-2020), methods seem appropriate, and data will be useful. However, some text could be provided about how these data might be useful beyond immediate application to hydrological modeling, for example, to biogeochemical modeling, for oxygen, organic carbon, redox-active species, contaminants, or nutrients? There are also a few missing references I identify below. The piezometer data are missing. Reference to those data should be removed entirely if they are not provided in the dataset.

**AC:**

We thank the referee for these comments. Here, we provide a point-by-point response and indicate where and how the manuscript was revised.

Data quality: I think the 10-minute and hourly averaged data should be provided, unless clearly justified why 10-minute data are not useful. I also think the code for the data processing should be provided for reproducibility. There is also no mention of data missingness, how missing data values were imputed or treated. A table showing the percentage of data coverage (completeness for each variable) might be helpful if there was significant data loss. And if data are imputed a binary flag column showing which rows were imputed could be helpful. In some cases, significant data were lost but there is no explanation. Why is there missing data from P2 for the first two months, and gaps from April to August in 2020 at P4? Finally, there is no header for the timestep variable in the Meteo text file, which creates a shift in variable names when the file is read into data analysis software.

As in response to most of the other reviewers, we now published all the time series data with the original 10-min resolution.

In the revised text, we added the information about the data loss in the new section 4 "Uncertainties and data usability".

Overall, 9.3 % of the soil water content data and 7.6 % of the soil temperature data are missing (particularly until March 2019 for the station North and between April and August 2020 for the station Valley) due to various technical failures.

We have corrected the headers in the meteo file in the newest data set version, thanks.

Line-by-line Comments

Line 32: I like this point about heterogeneity, but wouldn't processes be more localized if the steepest gradients were observed laterally?

Thanks for this comment. We rephrased the paragraph to avoid possible misunderstandings due to the use of the term "gradients".

Soils are widely distributed on the Earth surface. Flow and transport processes in unsaturated soils occur predominantly in the vertical direction, with the gravity force playing a major role, as abrupt changes in soil properties

due to soil horizons and layers are typically more significant than those in the lateral direction, and because of the strong coupling between soil, vegetation, and atmosphere. Therefore, despite the relevance of soils for global phenomena, the relevant soil processes are rather local.

Line 60: The citations Brantley et al., 2017 and White et al., 2015 (referring to CZOs) are missing from the references.

Thanks for spotting this. We have deleted the entire sentence about CZOs (and references therein) from the revised manuscript as it was somewhat out of the focus of the overall introduction section.

Line 64: Please state explicitly whether TERENO is part of the CZO network, or whether it is modeled after it?

See answer to the previous comment.

Figure 4: What is going on between April and mid-August 2020 for P4 and P5 soil water content? data are missing

Part of the SWC and soil temperature data measured between April and August 2020 went lost due to technical failures of the instrumentation installed in the valley bottom (i.e., profiles P4 and P5). We have now added this information to the revised manuscript in the new section 4 "Uncertainties and data usability".

Overall, 9.3 % of the soil water content data and 7.6 % of the soil temperature data are missing (particularly until March 2019 for the station North and between April and August 2020 for the station Valley) due to various technical failures.

Line 163: Why are groundwater data not shown? (only available after March 2020)
We have included the data in the revised manuscript as well as in the published data set.

Figure 7: Comment on similarity to other classification schemes, perhaps also include US soil order names in the text.

In the revised manuscript we have improved the soil description and referenced a previous work (Martini et al., 2015) where further information are provided.

Line 200: How was signal vs. noise determined?
In the revised manuscript we refer only to the full resolution data, hence we do not mention any hourly aggregation anymore.

**Authors' response to the referee comments RC6: "'Comment on essd-2020-363" by Anonymous Referee #6 (16 Feb 2021)**
* * *
**RC6:**
I'm yet another referee! I agree with most of the comments submitted by the referees and am glad to see the author's response to address them. The manuscript is written well and the dataset will be of interest to the hydrometeorological community. I just have a few points that I would like to be included in the revision.

**AC:**
We thank the referee for offering further recommendations, very useful from the modeling perspective.

1. The authors agreed to provide more information on the soil description, which is great. Figure 7 shows that texture class varies vertically at all stations, but it is difficult to extract the values as they overlap. It will be valuable to have a table for individual silt, sand, and clay % at each station for all depth, so different soil texture classes can be derived for different models. In addition, as a referee mentioned, dielectric permittivity, porosity, and soil bulk density are important parameters in the models that would be helpful to include as a list or reference.

The file "STH-net_Soils.txt" published as part of the data set includes the values of sand, silt, clay, porosity and bulk density for every soil profile and depth. This is also described in the manuscript. We think that providing the same information in the manuscript is beyond its scope.

2. Line 73 mentions that this experimental site aims to understand catchment-scale processes and the experimental site is a catchment size of 1.44 km2. Line 101 says the extent of the hillslope is 250x80 m, which seems to match the spatial maps (figure 1). I understand that P1-P8 are chosen according to the topographic gradient to capture the soil water dynamics along the transect of the gauging station in hillslope scale, but I think that it would be helpful to show where the hillslope is embedded within the catchment to get a bigger picture.

We edited Fig. 1 and added the map of the catchment with the position of the experimental hillslope site.

3. The datasets feature variables capturing soils heterogeneity; however, any additional information on vegetation and the site environment is helpful in a modeling perspective. The text mentions that the land cover is grass—can you add general description on how tall, growing season, time of the mowing, root depth, and LAI if available? The annual precipitation indicates it doesn't rain much, but does it snow and accumulate in winter?

In section 2 we state "this grassland transect is used as pasture and is not affected by agricultural practices except that the grass is mowed typically once per year".  More detailed information about the vegetation are not available, unfortunately.
We added a new section 4 "Uncertainties and data usability" to the revised manuscript where we also provide information about snow.

Until a few years ago, the Schäfertal catchment used to be affected by significant snowfall, with major snowmelt events occurring between January and April, whose effects on the hydrological processes are described, e.g., in Ollesch et al. (2005). In the last years, however, no significant snowfall events were observed. The last winter period (December 2020 to February 2021), instead, was characterized by exceptionally intense snowfall (with a maximum of ca. 45 cm on February 8th, 2021) that accumulated and persisted. Unfortunately, the technical infrastructure currently available at the site does not allow a meaningful estimation of the snow height and distribution during the monitoring period, hence the snowfall events are not recorded by the weather station in use (see Fig. 7). Because of this, the snow contribution to the water balance needs to be derived from the meteorological and soil temperature data available.